# Unraveling the Drifting Larval Fish Community in a Large Spawning Ground in the Middle Pearl River Using DNA Barcoding

**DOI:** 10.3390/ani12192555

**Published:** 2022-09-24

**Authors:** Weitao Chen, Ce Li, Xinhui Li, Jie Li, Yuefei Li

**Affiliations:** 1Pearl River Fisheries Research Institute, Chinese Academy of Fishery Sciences, Guangzhou 510380, China; 2Key Laboratory of Aquatic Animal Immune Technology of Guangdong Province, Guangzhou 510380, China; 3Guangzhou Scientific Observing and Experimental Station of National Fisheries Resources and Environment, Guangzhou 510380, China; 4Zhaoqing Aquatic technology Extension Center, Zhaoqing 526060, China

**Keywords:** Dongta spawning ground, molecular identification, species composition, fish stock

## Abstract

**Simple Summary:**

DNA barcoding identified 28 species from larvae collected from the Dongta spawning ground. Six invasive species were identified in the larval pool, implying that these species had successfully colonized the middle Pearl River. Several migratory species common in the lower Pearl River were rare or absent in the Dongta spawning ground, suggesting that adverse effects of dam construction posed on these migratory species. Our study provided important reference data for fishery management and conservation in the Pearl River.

**Abstract:**

Resolving the species composition of a larval pool in a spawning ground can provide novel insights into regional fish stocks and can support the development of effective monitoring and conservation policies. However, it is challenging to identify fish larvae to species due to their high diversity and dramatic phenotypic changes over development. In this study, we collected fish larvae in the Dongta spawning ground (Guiping City, Guangxi Province, China) in the middle reaches of the Pearl River between May and August 2018. We used a DNA barcoding approach to determine the species composition of the larval pool. A total of 905 larvae were chosen for molecular identification, of which 750 yielded high-quality barcoding sequences. Of these, 597 (≈79.6%), 151 (≈20.1%)/and 2 (≈0.3%) were assigned to 28 species, 8 genera, and 1 subfamily using the Barcode of Life Data System and GenBank nucleotide databases, respectively. Among the 28 identified species, 21 were cyprinids. Two species (*Mugilogobius myxodermus* and *Pseudolaubuca engraulis*) that were present only infrequently in previous adult surveys were abundant in the larval pool. Six invasive species were identified in the larval pool, implying that these species had successfully colonized the studied river section. Several migratory species common in the lower Pearl River were rare or absent in the investigated region, suggesting that dam construction in the Pearl River has had adverse effects on these migratory species. In summary, our study confirmed the applicability of DNA barcoding to studies of fish larval ecology and provided important reference data for fishery management and conservation in the Pearl River.

## 1. **Introduction**

Spawning grounds, which are centralized locations for fish spawning, play an ecologically significant role in fish reproduction [1,2,3,4,5]. However, spawning grounds are highly sensitive to environmental change, and ongoing anthropogenic factors, such as water engineering and pollution, have adversely affected fish spawning grounds worldwide [6,7,8,9,10,11]. Therefore, fish spawning grounds should be appropriately managed and protected globally. To support protective efforts, it is necessary to determine the current status of spawning grounds.

The Pearl River is the second largest drainage in China. This river system is famous for its rich fish diversity and high proportion of endemic species [12,13]. The Dongta river section, located in Guiping City, Guangxi Province, China, is an important spawning ground in the Pearl River due to its especial geographical conditions (Figure 1). Three large river branches meet in the Dongta section, increasing the richness of fish diversity. Notably, the Dongta was once the largest spawning ground for the four common Chinese carp species in the Pearl River [14]. Accordingly, this river section plays a key role in maintaining fish resources in the middle and lower Pearl River. Previous studies of the Dongta section mainly focused on measuring physical and chemical indices and assessing the reproductive hydrological requirement of individual species [14,15]; few reports have investigated which species finish spawning in this section.

Fish larvae provide important information about the early phases of fish development, conveying valuable data about reproduction and species composition in the spawning grounds. These data have important implications for the management of regional fish resources [3,16,17,18,19]. However, the morphological identification of some larval fishes to species level is notoriously difficult because fish larvae have limited diagnostic traits that can be used for identification [20,21]. Additionally, basic larval descriptions are lacking for numerous species, and even rare experts can identify fish larvae based on morphological characters. The difficulty of identifying fish larvae to species hinders the study of larval ecology.

DNA barcoding, a molecular identification approach based on the mitochondrial cytochrome *c* oxidase subunit I (COI) gene [22,23], has increasingly been used to identify fish larvae. DNA barcoding has provided new insights into the composition of larval fish communities [24,25,26,27,28]. Barcoding can identify larvae to species level in the absence of diagnostic morphological features. In this study, we used DNA barcoding to characterize the species composition of the larval fish community in the Dongta spawning ground and to investigate fish stocks from a larval perspective. Sampling was performed in 2018, during the main reproductive periods in the middle Pearl River (May–August).

## 2. Materials and Methods 

### 2.1. Collection of Fish Larvae

We sampled fish larvae in the Dongta spawning ground (Guiping City, Guangxi Province, China; 23.392° N, 110.089° E) between May and August 2018 (Figure 1). Samples were collected at intervals of 4–7 days, and each sampling period lasted for two hours. Sampling was performed at the following times: 8:00–10:00, 13:00–15:00, or 18:00–20:00. Fish larvae were collected using a customized net (total length, 2 m; rectangular iron opening/mouth, 1.0 m × 1.5 m; mesh size, 0.5 mm) attached to a filtrate collection bucket (0.8 m × 0.4 m × 0.4 m) and anchored parallel to the shoreline 10 m from shore. A total of 2985 fish larvae were handpicked and stored in 99% ethanol.

### 2.2. DNA Extraction, Amplification, and Sequencing

We chose 905 larvae for molecular identification on the basis of slight differences in morphology (Appendix A). Total genomic DNA was extracted using the Axygen DNA Extraction Kit (Axygen Scientific, Union City, CA, USA), according to the manufacturer’s specifications. The standard fish barcoding gene fragment (approximate 655-bp) was amplified using the universal fish primers FishF1 (5’-TCAACCAACCACAAAGACATTGGCAC-3’) and FishR1 (5’-TAGACTTCTGGGTGGCCAAAGAATCA-3’) [29]. The PCR cycling conditions were as follows: initial denaturation at 95 °C for 5 min; 30 cycles of 94 °C for 30 s, 54 °C for 30 s, and 72 °C for 1 min; and a final extension at 72 ℃ for 10 min. The PCR products were initially checked using 1.2% low-melting agarose gel electrophoresis and were subsequently sequenced bi-directionally on an ABI 3730XL DNA system (Perkin-Elmer Applied Biosystems, Foster City, CA, USA).

### 2.3. Data Analyses

The sequences were assembled on the basis of the high-quality tracer files using the Lasergene package (DNASTAR, Inc., Madison, WI, USA). The sequences were aligned using MUSCLE [30] and manually edited in MEGA version 6 [31]. The larval sequences were identified using the Barcode of Life Data System (BOLD [32]) as follows: We focused on the reference sequences of the best and second-best interspecific match in BOLD and documented the corresponding percentages of sequence matches. To represent species boundaries in the BOLD database, we adopted a 1% divergence threshold, as most species in the same region diverge less than 1% [33]. (i) If the unknown sequence was more than 99% similar to the best sequence match and was less than 99% similar to the second-best match, the sequence was unambiguously identified as the best-match species [25,34]. (ii) If the top 100 matches of the unknown sequence were the same species, a neighbor-joining (NJ) tree using the unknown larval sequences and matched sequences was constructed to testify the accuracy. The tree was produced in MEGA 6 with 1000 bootstrap replicates on the basis of the Kimura-2 parameter model [35]. (iii) If the unknown sequence was more than 99% or less than 99% similar to both the best match and the second-best match, and the two matches fell in the same genus, the unknown sequence was identified to genus level only. (iv) If the unknown sequence was more than 99% or less than 99% similar to both the best match and the second-best match, and the two matches fell in different genera, the unknown sequence was identified to subfamily or family level only. (v) If the unknown sequence did not match any reference sequences, these sequences were compared to the GenBank nucleotide database using the Basic Local Alignment Search Tool (BLAST; [36]). Finally, we then used MEGA 6 to build a neighbor-joining tree for identified larvae with 1000 bootstrap replicates based on the Kimura-2 parameter model to verify the accuracy of our identifications.

## 3. Results

### 3.1. Sequence Information

Of the 905 larvae selected for DNA barcoding, 832 were successfully amplified (91.9%). On the basis of the graph of sequence peaks, a total of 750 high-quality sequences were obtained (Appendix A). Due to poor sample preservation and/or primer specificity, 17.1% of the larvae (*n* = 155) did not yield PCR products or high-quality sequences. After alignment and trimming of noisy sites, we obtained 647 bp DNA barcodes. All sequences were longer than 600 bp, without stop codons or insertions, indicating that the collected sequences represented functional coding regions.

### 3.2. Molecular Identification of Fish Larvae

With respect to larval sequences that met criteria ii (Appendix A), the NJ tree revealed that all larval sequences were clustered with targeted species (Appendix A), suggesting the accurate identification of the larvae. Three larval sequences remained unidentified in the BOLD database can be delimited to genus level using GenBank nucleotide database (Appendix A). Therefore, according to the criteria defined above, 597 larval sequences (≈79.6%) were successfully assigned to 28 species, 151 sequences (≈20.1%) were assigned to eight genera, and 2 sequences were assigned to a single subfamily (Table 1, Appendix A). The identified larva fell into nine families and 32 genera (Table 1). The NJ tree showed that the identified species represented 45 independent lineages (Figure 2), suggesting that the larval samples included at least 45 distinct species and that the BOLD-based identifications were effective and accurate. Three and six putative species were detected in genera *Oreochromis* and *Rhinogobius*, respectively (Table 1, Figure 2). Species richness was highest in June (35 species; Table 1) and lowest in August (15 species; Table 1).

The 597 larval sequences identified to species were assigned to 28 species belonging to 23 genera in seven families (Table 1). The family Cyprinidae was the most abundant of the identified families; we identified 21 species (476 individuals) of Cyprinidae in this study, which accounted for 75% of all diagnosed species. Surprisingly, six invasive species (*Coptodon zillii*, *Cirrhinus mrigala*, *Hypostomus* sp., and three *Oreochromis* species) were detected in the larval pool. Across the 28 species identified, five species (*Chanodichthys recurviceps*, *Hemiculter leucisculus,*
*Pseudohemiculter dispar*, *Mugilogobius myxodermus,* and *Siniperca scherzeri*) were present in all four months, suggesting that these species may be the dominant taxa in the studied river section (Table 1).

## 4. Discussion

During our sampling, we found that a large number of the larvae collected in the Guiping river section were fragmentary due to the high velocity of the water flow. Thus, many larvae could not be identified on the basis of morphological features. In this study, DNA barcodes identified 79.6% of the larval sequences to a species level with a 99% similarity threshold, illustrating the importance of DNA-barcode-based identifications in studies of larval communities. Across all samples, DNA barcoding allowed us to delimit 28 species in 25 genera and seven families. In total, 21 of the 28 identified species (75.0%) were cyprinids, suggesting that the cyprinids may be the dominant fish taxon in the middle Pearl River. The proportion of cyprinids identified here was similar to the proportions of cyprinids in larval pools from the lower Pearl River (73.0%; Appendix A) [24] and the lower Hongshui River of the upper mainstem of the Pearl River (71.4%; Appendix A) [37]. In addition, two field surveys in the middle and lower Pearl River between 2005 and 2018 (Appendix A) reported that cyprinids were the dominant taxon [38,39]. Furthermore, species diversity in the larval pool peaked in June, similar to larval pools in previous studies [24,37]. Consequently, to better protect fish resources in the middle and lower Pearl River, the month of June should be considered a conservation priority.

Five species (*Chanodichthys recurviceps*, *Hemiculter leucisculus,*
*Pseudohemiculter dispar*, *Mugilogobius myxodermus,* and *Siniperca scherzeri*) were found in our samples at relatively high frequencies over all four months, suggesting that these species are unambiguously dominant in the middle Pearl River. With the exception of *Mugilogobius myxodermus*, the remaining four species have historically been common in the middle and lower reaches of the Pearl River [13,39,40] and/or in the field surveys that we have conducted over the past five years (data not published). However, the ubiquity of *Mugilogobius myxodermus* and *Pseudolaubuca engraulis* was unexpected, as these two species have been only infrequently present in recent field studies of the middle and lower Pearl River [13,39,40]; our annual field surveys). *Mugilogobius myxodermus* and *Pseudolaubuca engraulis* were also identified in the larval pool of the lower Pearl River [24], implying that these two species might represent relatively rich resources in the middle and lower Pearl River. *Mugilogobius myxodermus* and *Pseudolaubuca engraulis* are small, non-economic species and may be ignored by local fishermen, who primarily focus on economic species. In addition, these species are morphologically similar to many other small species and may have been misidentified in previous surveys. Of the species previously reported in the middle and lower Pearl River, at least three species (*Pseudolaubuca sinensis*, *Pseudohemiculter dispar*, and *Hemiculter leucisculus*) are morphologically similar to *Pseudolaubuca engraulis*, and six species (in the Gobiidae) are morphologically similar to *Mugilogobius myxodermus* [39,40]. The DNA-barcoding delimitation method used in this study thus provides a novel perspective on larval stocks and may prompt a reevaluation of the larval populations of species that might be only infrequently present in this region as adults.

Worryingly, six invasive species (*Cirrhinus mrigala, Coptodon zillii,*
*Hypostomus* sp., and three *Oreochromis* species) were detected in the larval pool, suggesting that these species have successfully colonized the middle Pearl River. The pattern has also been reported in the larval pool of the lower Pearl River [24]. In addition, field surveys have shown that these species are common in fish harvests and have even become the dominant species in some river sections [39,41]. Invasive fish populations can have adverse impacts on native species, and invasive species have thus become a key ecological problem in the Pearl River [42,43,44]. Therefore, these six invasive species deserve further study in the Pearl River.

*Ctenopharyngodon idella*, *Hypophthalmichthys molitrix*, *Hypophthalmichthys nobilis,* and *Mylopharyngodon piceus* are the most important economic species in China and are characterized by spawning migrations [3,13]. These four carp species are well represented in the larval pool of the lower Pearl River [24]. However, *Ctenopharyngodon idella* and *Hypophthalmichthys nobilis* were not identified in the larval pool of the middle Pearl River, while *Hypophthalmichthys molitrix* and *Mylopharyngodon piceus* were relatively rare. Thus, the resources of these four species were low in the analyzed river section. Changzhou Dam, which was constructed in 2004 in Wuzhou City, Guangxi Province, China (Figure 1), interrupts the migratory routes of these species and may have led to the observed resource decline. A similar pattern was reported for the migratory species *Megalobrama terminalis*, which is one of the most economically important species in the middle and lower Pearl River [13]. Across all of our samples, we identified only a single *Megalobrama terminalis* larva. *Megalobrama terminalis* was historically abundant in the upper reaches of the Pearl River at Wuzhou, Guangxi Province, China (Figure 1). However, this species was rarely captured in our previous annual surveys of the middle reach. Dam construction may have hindered migration spawning, with consequent significant adverse effects on migratory species populations.

Across all of our samples, certain larvae could only be identified to genus or family. The absence of published barcodes for Pearl River fishes is an important explanation for these incomplete identifications. To date, only a single barcoding study of Pearl River fishes has been performed, and this study included only 78 species [24]. Many more DNA barcoding studies of fish in the Pearl River are necessary to support future attempts to identify larvae based on genetic data. Furthermore, the standard fish barcoding marker only contains approximate 650 base pairs with limited genetic polymorphisms, which influences the resolution of species assignment, especially for closely relative species. Therefore, using longer markers or combining more molecular markers will increase the resolution of species assignment. Because it is unrealistic to use traditional DNA barcoding approaches to analyze an entire larval pool containing thousands of individual larvae, in the future, we will utilize DNA metabarcoding methods [45,46] to better resolve species composition in the larval pool.

## 5. Conclusions

The morphological characteristics of most fish larvae are insufficiently diagnostic, rendering identifications based on morphological features extremely difficult. Therefore, we used molecular barcodes to clarify the species composition of the largest larval pool in the Pearl River and attempted to assess Pearl River fish stocks from a larval perspective. Across the 750 larvae successfully barcoded, 597, 151, and 2 larvae were assigned to 28 species, 8 genera and 1 subfamily, respectively. Worryingly, 6 of the 28 identified species were invasive, and several traditionally abundant migratory species were rare or absent.

## Figures and Tables

**Figure 1 animals-12-02555-f001:**
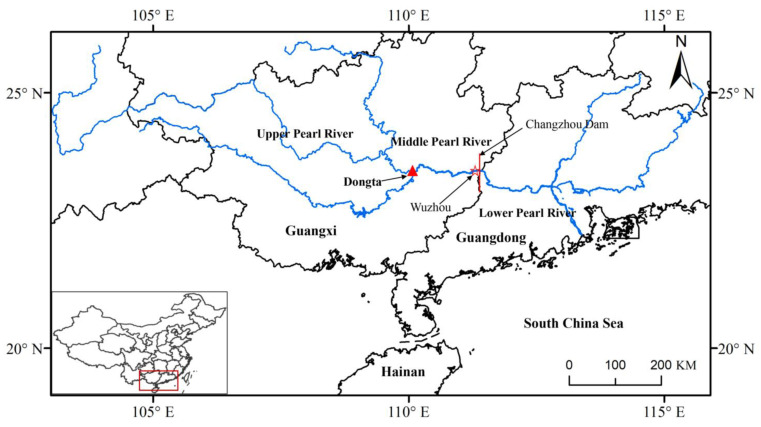
Map of the localities where fish larvae were collected in this study (triangle). Red line indicates the dams constructed in the middle and lower Pearl River.

**Figure 2 animals-12-02555-f002:**
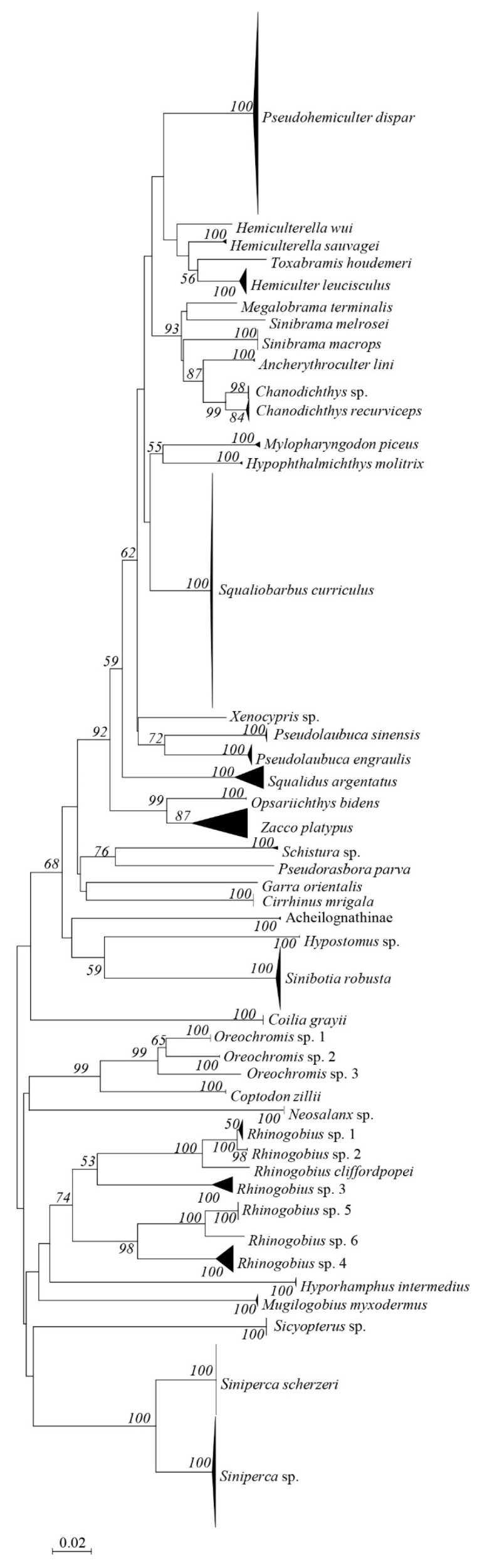
Neighbor-joining tree of 750 larvae. Bootstrap values are given at the nodes.

**Table 1 animals-12-02555-t001:** The community composition and occurrence times of taxa identified between May and August using DNA barcoding. N, the number of sequences for each species.

Family	Species	N	May	June	July	August
Botiidae	*Sinibotia robusta*	45		3	37	5
Nemacheilinae	*Schistura* sp.	2		1		1
Cichlidae	*Coptodon zillii*	3	1	1	1	
Cichlidae	*Oreochromis* sp. 1	5	2	2	1	
Cichlidae	*Oreochromis* sp. 2	2			2	
Cichlidae	*Oreochromis* sp. 3	1		1		
Cyprinidae	Acheilognathinae	2		1	1	
Cyprinidae	*Ancherythroculter lini*	2				2
Cyprinidae	*Cirrhinus mrigala*	9			9	
Cyprinidae	*Chanodichthys recurviceps*	43	7	6	26	4
Cyprinidae	*Chanodichthys* sp.	3		1	1	1
Cyprinidae	*Garra orientalis*	1			1	
Cyprinidae	*Hemiculter leucisculus*	18	8	8	1	1
Cyprinidae	*Hemiculterella sauvagei*	3	1	1		1
Cyprinidae	*Hemiculterella wui*	1		1		
Cyprinidae	*Hypophthalmichthys molitrix*	2			2	
Cyprinidae	*Megalobrama terminalis*	1	1			
Cyprinidae	*Mylopharyngodon piceus*	4		1	3	
Cyprinidae	*Opsariichthys bidens*	2	1	1		
Cyprinidae	*Pseudohemiculter dispar*	145	70	34	27	14
Cyprinidae	*Pseudolaubuca engraulis*	13	2	7	4	
Cyprinidae	*Pseudolaubuca sinensis*	10	1	2	5	2
Cyprinidae	*Pseudorasbora parva*	1		1		
Cyprinidae	*Sinibrama macrops*	15	1	11	3	
Cyprinidae	*Sinibrama melrosei*	1		1		
Cyprinidae	*Squalidus argentatus*	16	4	5	7	
Cyprinidae	*Squaliobarbus curriculus*	167		40	80	47
Cyprinidae	*Toxabramis houdemeri*	1		1		
Cyprinidae	*Xenocypris* sp.	1			1	
Cyprinidae	*Zacco platypus*	21	12	9		
Engraulidae	*Coilia grayii*	7	5	2		
Gobiidae	*Mugilogobius myxodermus*	8	3	2	1	2
Gobiidae	*Rhinogobius cliffordpopei*	1		1		
Gobiidae	*Rhinogobius* sp. 1	14	7	4	3	
Gobiidae	*Rhinogobius* sp. 2	2	2			
Gobiidae	*Rhinogobius* sp. 3	11	2	5	4	
Gobiidae	*Rhinogobius* sp. 4	19	3	8	2	6
Gobiidae	*Rhinogobius* sp. 5	2		2		
Gobiidae	*Rhinogobius* sp. 6	1	1			
Gobiidae	*Sicyopterus* sp.	2	1	1		
Hemiramphidae	*Hyporhamphus intermedius*	7	5	1	1	
Loricariidae	*Hypostomus* sp.	2			2	
Salangidae	*Neosalanx* sp.	6		2		4
Serranidae	*Siniperca* sp.	78	20	14	20	24
Serranidae	*Siniperca scherzeri*	50	17	13	11	9
Number of larvae		750	177	194	256	123
Number of species		45	24	35	27	15

## Data Availability

DNA sequences have been deposited in GenBank under accession numbers OP050461–OP051057. Details regarding individual samples are available in Appendix A.

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
