# Peer review of "Unraveling the Drifting Larval Fish Community in a Large Spawning Ground in the Middle Pearl River Using DNA Barcoding"

_animals, 2022, doi:10.3390/ani12192555_

Round 1
Reviewer 1 Report
General comments
The authors reveal the larval fish community in the middle Pearl River using DNA barcoding. This will be an important knowledge for the management and conservation of fisheries resources in the study area. In addition, the vast amount of fish DNA information obtained and accumulated has the potential to be used as basic information for future research on fish communities in the survey site. However, there are some uncertainties in the methodological description. The authors would need to revise their manuscript to eliminate these methodological ambiguities and make it more understandable. Thus, the substantial revision is needed to make this manuscript suitable for publication.
Specific comments
Methods
L96: FishF1 and FishR1 [29]
If possible, please add the approximate sequence length of PCR product, and the primer sequence for each primer.
L118–120: (iv) if the unknown sequence~
How do authors take into account the possibility of sequencing errors as well as an unknown species?
Or are sequencing errors checked as part of the process?
Results
L134–143: According to the criteria defined above, ~
The authors carry out collection surveys by time of day in each season.
However, in Results, the authors make comparisons of species richness among months, but do not mention comparisons between time periods. If possible, please add a comparison between time periods.
L144–145: The family Cyprinidae was the most abundant, including 20 genera identified in this study.
A statement of similar meaning appears in a later sentence (L146–148).
What does this sentence represent?
Discussion
L174–176: These field surveys identified 48 cyprinids (out of 96 total species) and 46 cyprinids~
A Table that allows comparison of the results of this study with those of previous studies would be helpful to the reader's understanding (e.g., additional descriptions to Table 1 or new Supplementary Table, etc.).
L228–L234: Across all of our samples, certain larvae could~
I also agree with authors about the difficulty of species identification due to lack of species in DNA barcode database. On the other hand, I think that the resolution of species assignment in the COI barcode region and the accumulation of information on genetic polymorphisms needs to be discussed.
(Although the number of target species may be small…)
L243–246: Because it is unrealistic to use traditional DNA~
Environmental DNA methods (eDNA metabarcoding) have recently begun to be used as a monitoring method for fish communities. Since many eDNA studies on monitoring fish communities have already been reported, eDNA methods may be useful in the authors' research.
Table1
In the species name, spp. is written in italics, please correct them.
Table S1
The accession number listed for each species could not be found in NCBI Search.
Have DNA sequencing information already been available?
In the species name, spp. is written in italics, please correct them.
Author Response
Comments and Suggestions for Authors
General comments
The authors reveal the larval fish community in the middle Pearl River using DNA barcoding. This will be an important knowledge for the management and conservation of fisheries resources in the study area. In addition, the vast amount of fish DNA information obtained and accumulated has the potential to be used as basic information for future research on fish communities in the survey site. However, there are some uncertainties in the methodological description. The authors would need to revise their manuscript to eliminate these methodological ambiguities and make it more understandable. Thus, the substantial revision is needed to make this manuscript suitable for publication.
Reply: Thanks for your valuable comments and suggestions. We will revise our MS point by point and highlighted in the submitted version.
Specific comments
Methods
L96: FishF1 and FishR1 [29]
If possible, please add the approximate sequence length of PCR product, and the primer sequence for each primer.
Reply: Done. (lines 110-113, page 5)
L118–120: (iv) if the unknown sequence~
How do authors take into account the possibility of sequencing errors as well as an unknown species?
Or are sequencing errors checked as part of the process?
Reply: First, we have checked the peaks of each sequence and kept high quality sequences. Second, we have conducted Blast search in NCBI for the unknown sequences and found that these sequences belong to fishes.
Results
L134–143: According to the criteria defined above, ~
The authors carry out collection surveys by time of day in each season.
However, in Results, the authors make comparisons of species richness among months, but do not mention comparisons between time periods. If possible, please add a comparison between time periods.
Reply: Though we have collected three times in each day, we have mixed them together for very few larvae were collected in each sample with only two hours and so we choose them based on morphological differences. Hence, we cannot conduct comparison between time periods.
L144–145: The family Cyprinidae was the most abundant, including 20 genera identified in this study.
A statement of similar meaning appears in a later sentence (L146–148).
What does this sentence represent?
Reply: Sorry for our careless and we have deleted one sentence in the revised MS.
Discussion
L174–176: These field surveys identified 48 cyprinids (out of 96 total species) and 46 cyprinids~
A Table that allows comparison of the results of this study with those of previous studies would be helpful to the reader's understanding (e.g., additional descriptions to Table 1 or new Supplementary Table, etc.).
Reply: We have made a table in supplementary (see below) .
|
|
Total species |
Cyprinidae species |
Ratio |
|
Dongta Spawn ground |
28 |
21 |
75.0% |
|
Larval pool in Lower Hongshui River [1] |
14 |
10 |
71.4% |
|
Larval pool in Lower Pearl River [2] |
37 |
27 |
73.0% |
|
Field survey in 2005 [3] |
99 |
46 |
46.5% |
|
Field survey in 2018 [4] |
96 |
48 |
50.0% |
L228–L234: Across all of our samples, certain larvae could~
I also agree with authors about the difficulty of species identification due to lack of species in DNA barcode database. On the other hand, I think that the resolution of species assignment in the COI barcode region and the accumulation of information on genetic polymorphisms needs to be discussed.
(Although the number of target species may be small…)
Reply: We have added this contents in the revised MS. (lines 258-262, page 10)
L243–246: Because it is unrealistic to use traditional DNA~
Environmental DNA methods (eDNA metabarcoding) have recently begun to be used as a monitoring method for fish communities. Since many eDNA studies on monitoring fish communities have already been reported, eDNA methods may be useful in the authors' research.
Reply: Thanks for your suggestions. We are now using eDNA metabarcoding methods to infer larval community in many river sections. We have revised the sentence in the revised paper. (lines 262-265, page 10)
Table1
In the species name, spp. is written in italics, please correct them.
Reply: we have corrected in all files.
Table S1
The accession number listed for each species could not be found in NCBI Search.
Have DNA sequencing information already been available?
In the species name, spp. is written in italics, please correct them.
Reply: We only submitted sequences to GenBank dataset, the accession number of sequences has not released now for we only submitted the sequences a month ago. But they will be released soon after. Furthermore, we have corrected spp. in the revised table S1.

Reviewer 2 Report
I have reviewed the MS by Weitao Chen et al. The authors sequenced larval fish DNA collected from the middle reach of the Pearl River. Samplings were conducted from May to August in 2018. In total 597 sequences were successfully obtained and assigned to 28 species. Although the study is not novel and the manuscript is almost exclusively descriptive, it would be of some significance to describe the basic information of species distribution.
I have a major concern on the method to assign the species name to the each sequence. The criteria (ii) (Lines 111-113) and the followings are not appropriate. There is no scientific justification for the use of the top 100 matches as the basis for species identification. If species identification cannot be done using the criteria (i), a phylogenetic tree should be created to identify the species.
Also, the authors only used BOLD database, but if the authors use the global BLAST, they can obtain some information on the unidentified species. It should be tested.
More minor concerns are as follows:
Line 13: The geographical information on the Dongta spawning ground should be described at the first appearance.
Figure 1: The letters in the figure should be enlarged.
Lines 92-100: The estimated amplicon length should be described here.
Lines 102-103: I cannot understand what was done here. What was spliced? The use of the word "contig" is not appropriate here.
Figure 3: This figure is not meaningful. Table is better to present these information.
Table 1. What is "n"? The number of species in each month should be described instead of "+".
Line 163: Authors did not perform "annual" sampling. Only four months.
Lines 199-200: Why is the Table 1 cited here?
Lines 243-246: The final sentence in the "Conclusions" is not conclusion. This statement should be moved to "Discussion".
Table S1: The arrangement of the samples are odd. For example, SZ2 appears after SZ19. Also, SZ1, SZ10, and SZ11 are duplicately presented in the end of the Table.
Tables S2 and S3: These tables are not cited in the text.
Author Response
Comments and Suggestions for Authors
I have reviewed the MS by Weitao Chen et al. The authors sequenced larval fish DNA collected from the middle reach of the Pearl River. Samplings were conducted from May to August in 2018. In total 597 sequences were successfully obtained and assigned to 28 species. Although the study is not novel and the manuscript is almost exclusively descriptive, it would be of some significance to describe the basic information of species distribution.
Reply: Reply: Thanks for your valuable comments and suggestions. We will revise our MS point by point and highlighted in the submitted version.
I have a major concern on the method to assign the species name to the each sequence. The criteria (ii) (Lines 111-113) and the followings are not appropriate. There is no scientific justification for the use of the top 100 matches as the basis for species identification. If species identification cannot be done using the criteria (i), a phylogenetic tree should be created to identify the species.
Reply: Thanks for your comments. Aiming to the criteria (ii), we have followed your suggestions and built a neighbor-joining (NJ) tree using the unknown sequences and matched sequences to testify the accuracy (see Fig. S1).
Also, the authors only used BOLD database, but if the authors use the global BLAST, they can obtain some information on the unidentified species. It should be tested.
Reply: Thanks for your suggestions. We have used the global BLAST search in NCBI and found that the unidentified species can be assigned in genus level (See Table S3). Furthermore, we have revised the results in the revised MS.
More minor concerns are as follows:
Line 13: The geographical information on the Dongta spawning ground should be described at the first appearance.
Reply: Added in the abstract section. (line 32, page 2)
Figure 1: The letters in the figure should be enlarged.
Reply: We have enlarged the letter in the revised figure 1.
Lines 92-100: The estimated amplicon length should be described here.
Reply: Added. (lines 110-113, page 5)
Lines 102-103: I cannot understand what was done here. What was spliced? The use of the word "contig" is not appropriate here.
Reply: Revised. (lines 120-121, page 5)
Figure 3: This figure is not meaningful. Table is better to present these information.
Reply: We have deleted this figure and added the information in Table 1.
|
Family |
Species |
N |
May |
June |
July |
August |
|
Botiidae |
Sinibotia robusta |
45 |
|
3 |
37 |
5 |
|
Nemacheilinae |
Schistura spp |
2 |
|
1 |
|
1 |
|
Cichlidae |
Coptodon zillii |
3 |
1 |
1 |
1 |
|
|
Cichlidae |
Oreochromis spp 1 |
5 |
2 |
2 |
1 |
|
|
Cichlidae |
Oreochromis spp 2 |
2 |
|
|
2 |
|
|
Cichlidae |
Oreochromis spp 3 |
1 |
|
1 |
|
|
|
Cyprinidae |
Acheilognathinae |
2 |
|
1 |
1 |
|
|
Cyprinidae |
Ancherythroculter lini |
2 |
|
|
|
2 |
|
Cyprinidae |
Cirrhinus mrigala |
9 |
|
|
9 |
|
|
Cyprinidae |
Chanodichthys recurviceps |
43 |
7 |
6 |
26 |
4 |
|
Cyprinidae |
Chanodichthys spp |
3 |
|
1 |
1 |
1 |
|
Cyprinidae |
Garra orientalis |
1 |
|
|
1 |
|
|
Cyprinidae |
Hemiculter leucisculus |
18 |
8 |
8 |
1 |
1 |
|
Cyprinidae |
Hemiculterella sauvagei |
3 |
1 |
1 |
|
1 |
|
Cyprinidae |
Hemiculterella wui |
1 |
|
1 |
|
|
|
Cyprinidae |
Hypophthalmichthys molitrix |
2 |
|
|
2 |
|
|
Cyprinidae |
Megalobrama terminalis |
1 |
1 |
|
|
|
|
Cyprinidae |
Mylopharyngodon piceus |
4 |
|
1 |
3 |
|
|
Cyprinidae |
Opsariichthys bidens |
2 |
1 |
1 |
|
|
|
Cyprinidae |
Pseudohemiculter dispar |
145 |
70 |
34 |
27 |
14 |
|
Cyprinidae |
Pseudolaubuca engraulis |
13 |
2 |
7 |
4 |
|
|
Cyprinidae |
Pseudolaubuca sinensis |
10 |
1 |
2 |
5 |
2 |
|
Cyprinidae |
Pseudorasbora parva |
1 |
|
1 |
|
|
|
Cyprinidae |
Sinibrama macrops |
15 |
1 |
11 |
3 |
|
|
Cyprinidae |
Sinibrama melrosei |
1 |
|
1 |
|
|
|
Cyprinidae |
Squalidus argentatus |
16 |
4 |
5 |
7 |
|
|
Cyprinidae |
Squaliobarbus curriculus |
167 |
|
40 |
80 |
47 |
|
Cyprinidae |
Toxabramis houdemeri |
1 |
|
1 |
|
|
|
Cyprinidae |
Xenocypris spp |
1 |
|
|
1 |
|
|
Cyprinidae |
Zacco platypus |
21 |
12 |
9 |
|
|
|
Engraulidae |
Coilia grayii |
7 |
5 |
2 |
|
|
|
Gobiidae |
Mugilogobius myxodermus |
8 |
3 |
2 |
1 |
2 |
|
Gobiidae |
Rhinogobius cliffordpopei |
1 |
|
1 |
|
|
|
Gobiidae |
Rhinogobius spp 1 |
14 |
7 |
4 |
3 |
|
|
Gobiidae |
Rhinogobius spp 2 |
2 |
2 |
|
|
|
|
Gobiidae |
Rhinogobius spp 3 |
11 |
2 |
5 |
4 |
|
|
Gobiidae |
Rhinogobius spp 4 |
19 |
3 |
8 |
2 |
6 |
|
Gobiidae |
Rhinogobius spp 5 |
2 |
|
2 |
|
|
|
Gobiidae |
Rhinogobius spp 6 |
1 |
1 |
|
|
|
|
Gobiidae |
Sicyopterus spp |
2 |
1 |
1 |
|
|
|
Hemiramphidae |
Hyporhamphus intermedius |
7 |
5 |
1 |
1 |
|
|
Loricariidae |
Hypostomus spp |
2 |
|
|
2 |
|
|
Salangidae |
Neosalanx spp |
6 |
|
2 |
|
4 |
|
Serranidae |
Siniperca spp |
78 |
20 |
14 |
20 |
24 |
|
Serranidae |
Siniperca scherzeri |
50 |
17 |
13 |
11 |
9 |
|
Number of larvae |
|
750 |
177 |
194 |
256 |
123 |
|
Number of species |
|
45 |
24 |
35 |
27 |
15 |
Table 1. What is "n"? The number of species in each month should be described instead of "+".
Reply: We have deleted the column including n. furthermore, we have used number instead of + in the revised MS. (see Table 1)
Line 163: Authors did not perform "annual" sampling. Only four months.
Reply: Revised. (line 183, page 7)
Lines 199-200: Why is the Table 1 cited here?
Reply: Deleted the citation.
Lines 243-246: The final sentence in the "Conclusions" is not conclusion. This statement should be moved to "Discussion".
Reply: Done. (lines 262-265, page 10)
Table S1: The arrangement of the samples are odd. For example, SZ2 appears after SZ19. Also, SZ1, SZ10, and SZ11 are duplicately presented in the end of the Table.
Reply: Revised. (see Table S1)
Tables S2 and S3: These tables are not cited in the text.
Reply: Added the two tables in the text. (lines 157-164, pages 6-7 )

Round 2
Reviewer 2 Report
This is the second time review for the manuscript by Weitao Chen et al. The authors successfully revised the manuscript. I now have only minor concerns as below:
Abstract: In addition to sequences that could be identified to species, sequences that could only be identified to genus or subfamily should be mentioned in the abstract. The same is true for the Conclusions.
Table1: The name of subfamily should not be Italicized.
Table1: Why are sequences that could not be identified to species denoted as "spp"? I think to use "sp." is better because "spp." means multiple unidentified species.
Author Response
Comments and Suggestions for Authors
This is the second time review for the manuscript by Weitao Chen et al. The authors successfully revised the manuscript. I now have only minor concerns as below:
Reply: Thanks for your valuable comments and suggestions. We will revise our MS point by point and highlighted in the submitted version.
Abstract: In addition to sequences that could be identified to species, sequences that could only be identified to genus or subfamily should be mentioned in the abstract. The same is true for the Conclusions.
Reply: Done. (lines 35-39, page 2; lines 273-274, page 10)
Table1: The name of subfamily should not be Italicized.
Reply: Done.
Table1: Why are sequences that could not be identified to species denoted as "spp"? I think to use "sp." is better because "spp." means multiple unidentified species.
Reply: Done. We have corrected figure 2 and all tables in the revised MS.